# ADAPTIVE PROCEDURAL TASK GENERATION FOR HARD-EXPLORATION PROBLEMS

**Kuan Fang**
Stanford University
`kuanfang@stanford.edu`

**Yuke Zhu**
UT Austin & Nvidia
`yukez@cs.utexas.edu`

**Silvio Savarese**
Stanford University
`ssilvio@stanford.edu`

**Li Fei-Fei**
Stanford University
`feifeili@stanford.edu`

## ABSTRACT

We introduce Adaptive Procedural Task Generation (APT-Gen), an approach to progressively generate a sequence of tasks as curricula to facilitate reinforcement learning in hard-exploration problems. At the heart of our approach, a task generator learns to create tasks from a parameterized task space via a black-box procedural generation module. To enable curriculum learning in the absence of a direct indicator of learning progress, we propose to train the task generator by balancing the agent's performance in the generated tasks and the similarity to the target tasks. Through adversarial training, the task similarity is adaptively estimated by a task discriminator defined on the agent's experiences, allowing the generated tasks to approximate target tasks of unknown parameterization or outside of the predefined task space. Our experiments on grid world and robotic manipulation task domains show that APT-Gen achieves substantially better performance than various existing baselines by generating suitable tasks of rich variations.[1]

## 1 INTRODUCTION

The effectiveness of reinforcement learning (RL) relies on the agent's ability to explore the task environment and collect informative experiences. Given tasks handcrafted with human expertise, RL algorithms have achieved significant progress on solving sequential decision making problems in various domains such as game playing (Badia et al., 2020; Mnih et al., 2015) and robotics (OpenAI et al., 2019; Duan et al., 2016). However, in many hard-exploration problems (Aytar et al., 2018; Paine et al., 2020), such trial-and-error paradigms often suffer from sparse and deceptive rewards, stringent environment constraints, and large state and action spaces.

A plurality of exploration strategies has been developed to encourage the state coverage by an RL agent (Houthooft et al., 2016; Pathak et al., 2017; Burda et al., 2019; Conti et al., 2018). Although successes are achieved in goal-reaching tasks and games of small state spaces, harder tasks often require the agent to complete a series of sub-tasks without any positive feedback until the final mission is accomplished. Naively covering intermediate states can be insufficient for the agent to connect the dots and discover the final solution. In complicated tasks, it could also be difficult to visit diverse states by directly exploring in the given environment (Maillard et al., 2014).

In contrast, recent advances in curriculum learning (Bengio et al., 2009; Graves et al., 2017) aim to utilize similar but easier datasets or tasks to facilitate training. Being applied to RL, these techniques select tasks from a predefined set (Matiisen et al., 2019) or a parameterized space of goals and scenes (Held et al., 2018; Portelas et al., 2019; Racanière et al., 2020) to accelerate the performance improvement on the target task or the entire task space. However, the flexibility of their curricula is often limited to task spaces using low-dimensional parameters, where the search for a suitable task is relatively easy and the similarity between two tasks can be well defined.

---

[1]Project page: `https://kuanfang.github.io/apt-gen/`

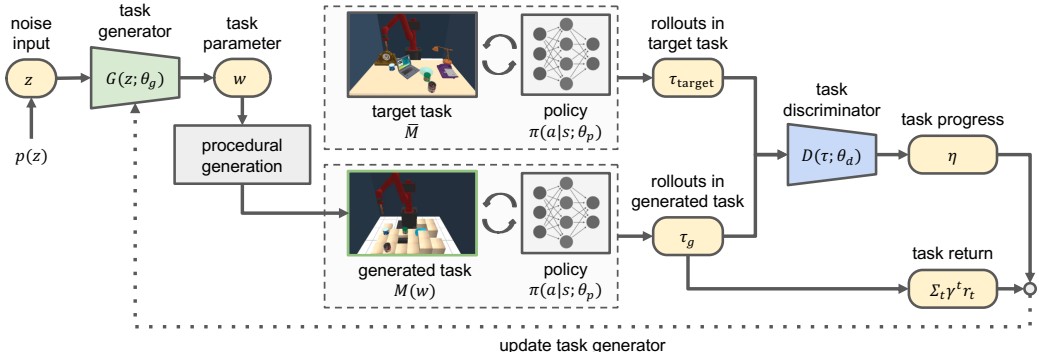

Figure 1: APT-Gen learns to create tasks via a black-box procedural generation module. By jointly training the task generator, the task discriminator, and the policy, suitable tasks are progressively generated to expedite reinforcement learning in hard-exploration problems.

In this work, we combat this challenge by generating tasks of rich variations as curricula using procedural content generation (PCG). Developed for automated creation of environments in physics simulations and video games (Summerville et al., 2018; Risi & Togelius, 2019; Cobbe et al., 2020), PCG tools have paved the way for generating diverse tasks of configurable scene layouts, object types, constraints, and objectives. To take advantage of PCG for automated curricula, the key challenge is to measure the learning progress in order to adaptively generate suitable tasks for efficiently learning to solve the target task. In hard-exploration problems, this challenge is intensified since the performance improvement cannot always be directly observed on the target task until it is close to being solved. In addition, the progress in a complex task space is hard to estimate when there does not exist a well-defined measure of task difficulty or similarity. We cannot always expect the agent to thoroughly investigate the task space and learn to solve all tasks therein, especially when the target task has unknown parameterization and the task space has rich variations.

To this end, we introduce Adaptive Procedural Task Generation (APT-Gen), an approach to progressively generate a sequence of tasks to expedite reinforcement learning in hard-exploration problems. As shown in Figure 1, APT-Gen uses a task generator to create tasks via a black-box procedural generation module. Through the interplay between the task generator and the policy, tasks are continuously generated to provide similar but easier scenarios for training the agent. In order to enable curriculum learning in the absence of a direct indicator of learning progress, we propose to train the task generator by balancing the agent's performance in the generated tasks and the task progress score which measures the similarity between the generated tasks and the target task. To encourage the generated tasks to require similar agent's behaviors with the target task, a task discriminator is adversarially trained to estimate the task progress by comparing the agent's experiences collected from both task sources. APT-Gen can thus be trained for target tasks of unknown parameterization or even outside of the task space defined by the procedural generation module, which expands the scope of its application. By jointly training the task generator, the task discriminator, and the policy, APT-Gen is able to adaptively generate suitable tasks from highly configurable task spaces to facilitate the learning process for challenging target tasks.

Our experiments are conducted on various tasks in the grid world and robotic manipulation domains. Tasks generated in these domains are parameterized by $6\times$ to $10\times$ independent variables compared to those in prior work (Wang et al., 2019; 2020; Portelas et al., 2019). Each task can have different environment layouts, object types, object positions, constraints, and reward functions. In challenging target tasks of sparse rewards and stringent constraints, APT-Gen substantially outperforms existing exploration and curriculum learning baselines by effectively generating new tasks during training.

## 2 RELATED WORK

**Hard-Exploration Problems.** Many RL algorithms aim to incentivize the agent to visit more diverse and higher-reward states. Methods on intrinsic motivation augment the sparse and deceptive environment rewards with an additional intrinsic reward that encourages curiosity (Pathak et al.,

2017; Burda et al., 2019; Raileanu & Rocktäschel, 2020) and state novelty (Conti et al., 2018; Eysenbach et al., 2019). Another family of exploration techniques is derived from an information-theoretical perspective as maximizing information gain of actions (Houthooft et al., 2016; Sun et al., 2011). When human demonstrations are available, they can be used to facilitate an RL agent to visit similar states and transitions as illustrated in the demonstrations (Vecerik et al., 2017; Nair et al., 2018; Zhu et al., 2018). A combination of these techniques has been applied to solve hard-exploration problems in video game domains (Aytar et al., 2018; Ecoffet et al., 2019). However, these methods have focused on learning in relatively simple and fixed environments, and usually can be ineffective in tasks where explorations are thwarted by stringent environment constraints or naively covering states does not lead to the task success.

**Curriculum Learning.** Curriculum learning utilizes alternative datasets and tasks to accelerate the learning process of challenging target tasks (Bengio et al., 2009; Graves et al., 2017). To apply curriculum learning to RL, several recent works learn to adaptively select a finite set of easy tasks (Narvekar et al., 2017; Svetlik et al., 2017; Riedmiller et al., 2018; Peng et al., 2018; Czarnecki et al., 2018; Matiisen et al., 2019; Narvekar & Stone, 2019; Lin et al., 2019) or auxiliary rewards (Jaderberg et al., 2017; Shen et al., 2019) hand-designed by human to maximize a progress signal defined on the target task. Parameterized tasks have been used to form a curriculum through the configuration of goals (Forestier et al., 2017; Held et al., 2018; Racanière et al., 2020), environment layouts (Wöhlke et al., 2020; Baker et al., 2020; Portelas et al., 2019), and reward functions (Gupta et al., 2018; Jabri et al., 2019). OpenAI et al. (2019) and Mehta et al. (2020) propose to actively adjust the hyperparameters in physical simulators to alleviate the domain shift. Most of these works are designed for task spaces parameterized by several discrete or continuous variables, where the task space can often be thoroughly explored and the similarity between two tasks could be well defined in the parameter space. In contrast, our approach is able to effectively generate tasks parameterized by a combination of high-dimensional discrete variables and several continuous variables. While most of these works focus on parameterizing a single aspect of the task environment, our approach learns to generate new tasks of rich variations with configurable initial state probability, transition probability, and reward function. Sukhbaatar et al. (2018b), Florensa et al. (2017), and Sukhbaatar et al. (2018a) propose to use an adversarial agent to set goals of growing difficulties by reversely traversing the state space from the goal. While this is related to the adversarial training framework in this paper in principle, we apply our framework beyond goal-reaching and reversible task domains.

**Procedural Task Generation.** Procedural generation has been widely used in computer graphics and robotics (Fisher et al., 2012; Izadinia et al., 2017; Majerowicz et al., 2013; Izatt & Tedrake, 2020; Schwarz & Behnke, 2020). While an increasing number of task sets have been designed to benchmark and empower reinforcement learning research (Kolve et al., 2017; Xia et al., 2018; Savva et al., 2019; Yu et al., 2019; James et al., 2020), the design and implementation of each task often require nontrivial human expertise and heavy engineering. A few recent works utilize the random procedural generation of tasks (Cobbe et al., 2020; Fang et al., 2018; Raileanu & Rocktäschel, 2020; Silver & Chitnis, 2020). However, their generation algorithms are handcrafted with limited configurable features. Evolution strategies (Wang et al., 2019; 2020), automated procedures (Justesen et al., 2018), and learning-based methods (Gravina et al., 2019; Khalifa et al., 2020; Bontrager & Togelius, 2020) have been proposed to automatically discover diverse games and task environments for training RL agents. Instead of covering the entire task space or discovering a diverse set of policies in an open-ended manner, our approach aims to train the policy to solve the target tasks of interest by utilizing the generated tasks.

## 3 ADAPTIVE PROCEDURAL TASK GENERATION

We consider a reinforcement learning problem involving a target task that the policy learns to solve and a parameterized task space that we utilize to generate new tasks. In practice, the parameterized task space can be created by a simulation program or a configurable procedure to set up the environment by a human or a robot in the real world. The target task can be an instance of an unknown parameter or a task outside of the task space, as long as there exist shared properties and transferable knowledge between the generated tasks and the target task. This follows the general paradigm of teacher-student curriculum learning (Matiisen et al., 2019; Portelas et al., 2019), while we allow the task space to be parameterized by either continuous or discrete high-dimensional variables and we do not assume the target task has a known parameterization by these variables.

We propose Adaptive Procedural Task Generation (APT-Gen), an approach for progressively generating tasks in highly configurable task spaces as curricula. To enable curriculum learning for hard-exploration problems, our key insight is that the learning progress can be jointly estimated by how well the policy can solve the current generated tasks and how similar the generated tasks are to the target task. Starting with a set of tasks that the policy can easily learn to solve, our approach progressively adapts the generated tasks towards the target task while maintaining their feasibility to the policy. As shown in Figure 1, our approach creates tasks via a black-box procedural generation module by jointly learning the task generator, the task discriminator, and the policy.

## 3.1 PROBLEM FORMULATION

We consider each task as a Markov Decision Process (MDP) denoted by a tuple $M = (\mathcal{S}, \mathcal{A}, \rho, P, R, \gamma)$ with state space $\mathcal{S}$, action space $\mathcal{A}$, initial state probability $\rho$, transition probability $P$, reward function $R$, and discount factor $\gamma$. The task space $\mathcal{T}$ defines a finite or infinite number of MDPs of similar designs and properties. We use a multi-dimensional parameter space $\mathcal{W}$ to represent the inter-task variation of $\mathcal{T}$. Given a task parameter $w \in \mathcal{W}$, a task $M(w)$ can be instantiated in the task space by a predefined mapping $M(\cdot)$. While a generic task space can be composed of fully configurable MDPs, in this work we assume that all tasks share the same $\mathcal{S}$, $\mathcal{A}$ and $\gamma$ such that all policies share the same input and output dimensions. In this case, each $M(w)$ is defined by a distinct set of $\rho$, $P$, $R$ parameterized by $w$. The target task $\overline{M}$ is either an instance of unknown parameter $\overline{w} \in \mathcal{W}$ or a task outside of $\mathcal{T}$ but shares the same $\mathcal{S}$ and $\mathcal{A}$.

Our goal is to learn a policy $\pi$ to solve the target task $\overline{M}$. During training, the curriculum is formed as a sequence of task parameters $\{w_i\}_{i=1}^{N}$ with index $i$ for constructing the corresponding sequence of generated tasks $M(w_i)$. The agent collects rollouts by unrolling in both $\overline{M}$ and $M(w_i)$. Each rollout is denoted as $\tau$, which is composed of a sequence of state $s_t$, action $a_t$ and reward $r_t$ at each time step $t$. In the generated tasks, the $w_i$ of the source task is recorded alongside with the $\tau$. Given a fixed budget of total collected steps in both task domains the objective is to maximize the policy's expected return $\mathbb{E}[\sum_t \gamma^t r_t]$ in the target task $\overline{M}$.

## 3.2 ADAPTIVE GENERATION FOR HARD-EXPLORATION PROBLEMS

The interplay between the policy $\pi(a|s; \theta_p)$ and the task generator $G(z; \theta_g)$ is formulated in a teacher-student paradigm (Matiisen et al., 2019), where $\theta_p$, $\theta_g$ are learnable model parameters and $z$ is a noise input used in deep generative models (Goodfellow et al., 2014). In contrast to prior work (Held et al., 2018; Matiisen et al., 2019; Portelas et al., 2019; Racanière et al., 2020) which rely on evaluating performance improvements directly on the target task or the entire task space, we propose to define the indicator of learning progress using the expected return $\mathbb{E}[\sum_t \gamma^t r_t]$ and a task progress $\eta$ to enable curriculum learning in hard-exploration problems. The expected return measures the policy's performance in the generated tasks sampled by $G$. While the task progress $\eta$ is a continuous score which represents the generated tasks' similarity to the target task. The definition and learning process of $\eta$ will be detailed in Sec. 3.3. When both the expected return and the task progress reach the maxima, the generated tasks are supposed to be indistinguishable from the target task and $\pi$ is trained to be the optimal policy for the target task.

The training requires a careful balance between the task progress and the expected return. A highly configurable task space potentially contains a large amount of tasks that are infeasible or of similar difficulties with the target task. If the task distribution of $G$ moves too fast towards the target task, the policy can quickly be overwhelmed by difficult tasks and lose track of what tasks can be effectively learned. On the contrary, sticking to the tasks that can be solved by the current policy will retard the learning progress and overfit the policy to the easy scenarios. Our approach maximizes the task progress subject to a target minimum expected return $\delta$ as a chosen hyperparameter. Then the training of the task generator amounts to the optimization problem:

$$\max_{\theta_g} \mathbb{E}_{w \sim G}[\eta], \qquad \text{subject to} \quad \mathbb{E}_{\tau \sim G, \pi}[\sum_t \gamma^t r_t] \geq \delta, \tag{1}$$

where $w \sim G$ represents the generation process jointly determined by $p(z)$ and $G$ and $\mathbb{E}_{\tau \sim G, \pi}[\cdot]$ is a shorthand notation for $\mathbb{E}_{w \sim G}[\mathbb{E}_{\tau \sim M(w), \pi}[\cdot]]$ to represent the expectation over distribution of

---

**Algorithm 1** Adaptive Procedural Task Generation (APT-Gen)

---

**Require:** target task $\overline{M}$, parameterized task space $M(\cdot)$, prior probability $p(z)$, learning rate $\alpha$
 1: Initialize parameters $\theta_p, \theta_g, \theta_d, \theta_1, \theta_2, \beta$
 2: Initialize replay buffers $\mathcal{D}_g$ and $\mathcal{D}_{target}$
 3: **while** not converged **do**
 4:     Sample $z \sim p(z)$ and create the generated task $M(w)$ with $w = G(z; \theta_g)$
 5:     Collect a rollout $\tau_g$ in $M(w)$ using $\pi(a|s; \theta_p)$ and store $w$ and $\tau_g$ in $\mathcal{D}_g$
 6:     Collect a rollout $\tau_{target}$ in $\overline{M}$ using $\pi(a|s; \theta_p)$ and store $\tau_{target}$ in $\mathcal{D}_{target}$
 7:     Update $\theta_d \leftarrow \theta_d - \alpha \nabla_{\theta_d} \mathcal{L}_d(\theta_d, \mathcal{D}_{target}, \mathcal{D}_g)$
 8:     Update $\theta_1 \leftarrow \theta_1 - \alpha \nabla_{\theta_1} \mathbb{E}_{w, \tau_g \sim \mathcal{D}_g}[(V_1(w; \theta_1) - D(\tau_g; \theta_d))^2]$
 9:     Update $\theta_2 \leftarrow \theta_2 - \alpha \nabla_{\theta_2} \mathbb{E}_{w, \tau_g \sim \mathcal{D}_g}[(V_2(w; \theta_2) - \sum_t \gamma^t r_t)^2]$
10:     Update $\theta_g \leftarrow \theta_g - \alpha \nabla_{\theta_g} \mathcal{L}_g(\theta_g, \theta_1, \theta_2, \beta)$
11:     Update $\beta$ as described in Sec. 3.2
12:     Update $\theta_p$ using the RL algorithm with sampled batches from $\mathcal{D}_g$ and $\mathcal{D}_{target}$
13: **end while**

---

rollouts. By re-writing this optimization problem as a Lagrangian, we obtain:

$$\max_{\theta_g} \left( \mathbb{E}_{w \sim G}[\eta] + \beta(\mathbb{E}_{\tau \sim G, \pi}[\sum_t \gamma^t r_t] - \delta) \right), \qquad (2)$$

where $\beta$ is the Lagrangian multiplier that balances the task feasibility and the task progress. $\beta$ can have a delayed effect on the optimization problem since $\pi$ and $\eta$ are learned at the same time. We adopt an automated procedure (Schulman et al., 2017) to adjust $\beta$ adaptively when $\mathbb{E}_{\tau \sim G, \pi}[\sum_t \gamma^t r_t] - \delta$ exceeds a threshold.

Since the gradients cannot be directly backpropagated to the task generator, we follow the practice of Konda & Tsitsiklis (2000); Matiisen et al. (2019) and use two value functions to estimate the two expectation terms respectively. By taking the input task parameter $w$, the progress value function $V_1(w; \theta_1)$ estimates the task progress $\eta$ and the return value function $V_2(w; \theta_2)$ estimates the expected return $\mathbb{E}_{\tau \sim M(w), \pi}[\sum_t \gamma^t r_t]$, where $\theta_1$ and $\theta_2$ are learnable model parameters. The two value functions are trained to fit the two expectation terms over the distribution of $\tau$ with respect to $\theta_1$ and $\theta_2$, using rollouts collected in the generated tasks with the policy $\pi$. The training of the task generator becomes learning $\theta_g$ to maximize the task generator loss:

$$\mathcal{L}_g(\theta_g, \theta_1, \theta_2, \beta) = \mathbb{E}_{z \sim p(z)}[V_1(G(z; \theta_g); \theta_1) + \beta(V_2(G(z; \theta_g); \theta_2) - \delta)]. \qquad (3)$$

### 3.3 Adversarial Training of Task Progress

The goal of the task progress $\eta$ is to guide the task generator $G$ to generate tasks similar to the target task. Since the difficulty level and the task similarity cannot be defined by an objective metric in many complex task domains, we argue that $\eta$ needs to jointly adapt with $G$ and $\pi$ when the task distribution and the policy constantly evolve over the course of training. Ideally, $\eta$ should satisfy two requirements: First, when the maximum $\eta$ is achieved at convergence, a generated task $M(w)$ and the target task $\overline{M}$ should be indistinguishable from the perspective of the policy $\pi$. Second, since a small change in an ill-posed task parameter space can completely alter the required agent's behaviors to solve the task, $\eta$ needs to provide a smooth signal to adapt $G$ in the task space.

To this end, we estimate $\eta$ using a task discriminator $D(\tau; \theta_d)$ defined on the agent's experiences in the task environment, where $\theta_d$ is the learnable model parameter. It takes $\tau$ as input and learns to estimate the probability of the task $M$ being the target task $\overline{M}$ conditioned on the rollout $\tau$ induced by the policy $\pi$. The task progress $\eta$ of the task parameter $w$ can be defined as $\mathbb{E}_{\tau \sim M(w), \pi}[D(\tau; \theta_d)]$. In this way, $D$ forms an adversarial training framework (Goodfellow et al., 2014) against $G$ and $\pi$, which jointly determine the likelihood of $\tau$.

The task discriminator is required to comprehensively compare the given task with the target task in APT-Gen. Unlike prior work which aims to discriminate policies (Ho & Ermon, 2016) and physics parameters (Mehta et al., 2020), $D$ computes the prediction score by taking the overall MDP definition into account. Therefore, $D$ is designed to separately encode the initial state $s_1$ and each transition $(s_t, a_t, r_t, s_{t+1})$ of step $t$ to discriminate the initial state probability $\rho$, the transition probability

$P$ and the reward function $R$ respectively. The prediction is computed using a pooling function across all encoded features. The implementation details of $D$ are described in Appendix B.

To train the task generator, we collect rollouts from generated tasks as $\tau_g$ and the target task as $\tau_{target}$, stored in two replay buffers $\mathcal{D}_g$ and $\mathcal{D}_{target}$ respectively. The training of $D(\tau; \theta_d)$ is conducted by minimizing a discriminator loss (Goodfellow et al., 2014) to classify the task sources of the collected rollouts:

$$\mathcal{L}_d(\theta_d, \mathcal{D}_{target}, \mathcal{D}_g) = -\mathbb{E}_{\tau_{target} \sim \mathcal{D}_{target}}[\log(D(\tau_{target}; \theta_d))] - \mathbb{E}_{\tau_g \sim \mathcal{D}_g}[1 - \log(D(\tau_g; \theta_d))]. \quad (4)$$

In principle, to learn a $G$ that produces the exact MDP definition of $\overline{M}$, we would require $\overline{M}$ to be an instance of the task space $\mathcal{T}$ and the training data to be collected by arbitrary $\pi$ to fully investigate differences in the two task environments. However, this could be neither computationally practical nor necessary. Given that our goal is to find an optimal policy $\pi^*$ to solve the target task, we only need $\overline{M}$ and $M(w)$ to be indistinguishable from the perspective of the policy. In practice, the rollouts are collected using the updated $\pi(a|s, w; \theta_p)$ with epsilon-greedy exploration (Sutton & Barto, 2011). One could also encourage explorations (Pathak et al., 2017) in the policy learning to efficiently distinguish the two tasks, which we leave out of the scope of this work.

The pseudocode of the algorithm is outlined in Algorithm 1. The training alternates among updates of the policy, the task discriminator, and the task generator in the same loop. New rollouts are continuously collected from both the target task and the generated tasks using the updated $\pi$. In this work, we equally collect experiences from the two sources to train the policy, while a smarter strategy of choosing between task sources can be further investigated in future work.

## 4 EXPERIMENTS

The goal of our experimental evaluation is to answer the following questions: 1) Can APT-Gen facilitate reinforcement learning in hard-exploration problems? 2) What tasks can be generated by APT-Gen for a target task during training? 3) Can APT-Gen be applied to target tasks outside of the task space predefined by the procedural generation module?

### 4.1 TASKS

The experiments are conducted in two configurable task spaces: `Grid-World` and `Manipulation`. The two task spaces are configured by 74 and 31 parameters consist of a combination of discrete and continuous variables. Each task space contains various tasks that share the same state and action spaces but different environment layouts, object types, object positions, constraints, and reward functions. The tasks from these task spaces can have stringent penalties and constraints such as lava regions and pitfalls which make it hard for the agent to succeed or survive. In contrast to many handcrafted hard-exploration tasks in prior work (Salimans & Chen, 2018) which provide sub-task rewards (e.g. finding a key and opening a door), our tasks only provide a

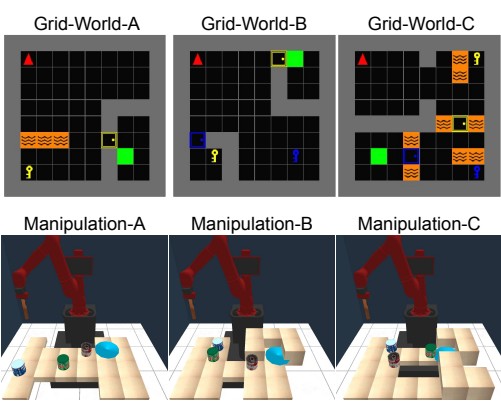

Figure 2: Target tasks in the two domains.

sparse positive reward when the final task is accomplished, which introduces extra challenges for the agent. As shown in Figure 2, we design three target tasks of different complexities in each task space for evaluation. Details of the task design and the task parameterization can be found in Appendix A.

### 4.2 QUANTITATIVE RESULTS

We evaluate the performance of the agent in target tasks using different methods. All methods are trained with a fixed budget of total steps collected by the agent. In Appendix B, we provide implementation details, hyperparameters, training and evaluation protocols.

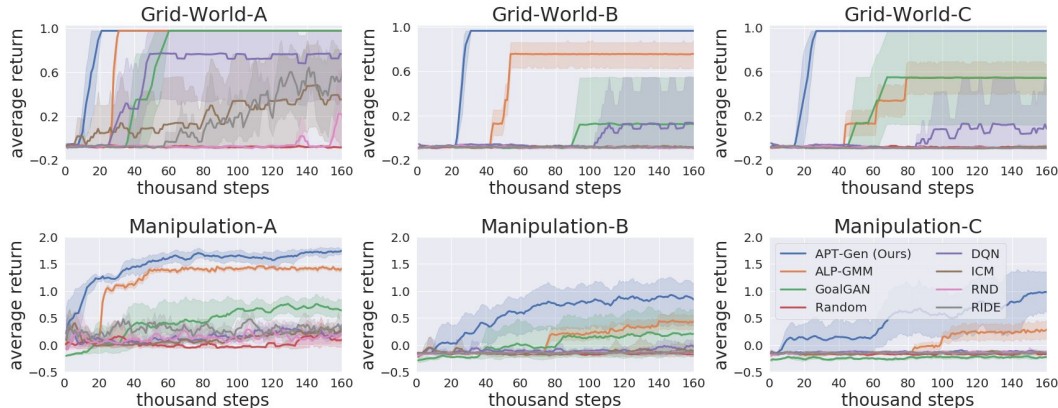

Figure 3: Quantitative results of the performance of the agent in the target tasks.

**Baselines.** We compare APT-Gen with various baselines, including 3 exploration strategies, 3 curriculum methods, and a model-free RL baseline. ICM (Pathak et al., 2017), RND (Burda et al., 2019), and RIDE (Raileanu & Rocktäschel, 2020) adversarially learn intrinsic motivations to encourage exploration in the state space. Random uniformly samples from the task parameter space to create tasks. ALP-GMM (Portelas et al., 2019) and GoalGAN (Held et al., 2018) use Gaussian Mixture Models and GAN to sample tasks as curricula without mechanisms to estimate the task distance and handle complex task spaces. DQN (Mnih et al., 2013; Hessel et al., 2018) directly applies Q-learning in the target task. To have a fair comparison, we use the same architecture for the corresponding components and search for the optimal hyperparameters for each method.

**Comparative Analysis.** In Figure 3, we present the agent's performance in the target tasks by using different methods. The average return across 5 independent runs is reported and the shaded area indicates the standard deviation. To have fair comparisons, the x-axes of APT-Gen and curriculum learning baselines (Random, GoalGAN, ALP-GMM) indicate the total steps collected from the target and generated tasks, while x-axes of other baselines (DQN, ICM, RND, RIDE) indicate the steps collected from only the target task. In all scenarios, our approach achieves superior performance comparing with baseline methods. In `Grid-World` tasks, APT-Gen successfully trains the agent to find keys in separate locations and access different rooms in the right sequential order. In the `Manipulation` tasks, APT-Gen enables the agent to solve the puzzle by moving around the obstacles in the correct order without causing collisions. Especially, in `Manipulation-C`, our agent develops an effective strategy that first moves away from the target object away to yield the path for the obstacle to leave and then pushing it back towards the goal to complete the task.

Most baseline methods fail in hard tasks that require sequential problem solving over a longer horizon, although some can achieve comparable results in easier scenarios (i.e. when there is only one room and the environment is mostly empty). ICM, RND, and RIDE demonstrate effective explorations when the environment is relatively simple, but the agent is often thwarted by the penalties caused by constraints in the environment. Without any reward shaping for sub-tasks, naively reaching to the intermediate states (e.g. finding the keys) does not yield any immediate reward unless the goal is reached at the end of the same episode. Without mechanisms to estimate the task similarity and to handle complex task spaces, curriculum learning baselines like ALP-GMM and GoalGAN fail to produce useful tasks that share similar challenges with the target tasks.

**Out-of-Space Task.** To demonstrate APT-Gen's performance in target tasks that are outside of the predefined task space, we train the model to solve a different robotic manipulation task while still generating tasks from the task space defined in Sec. 4.1. The target task shares the same state and action spaces with the predefined task space, but the table has a different shape and a variety of static objects are placed on the table as environment constraints. As shown in Figure 4, APT-Gen efficiently learns to solve the out-of-space task while baseline methods take much more steps or completely fail to learn. Qualitative results of out-of-space tasks will be discussed in Sec. 4.4.

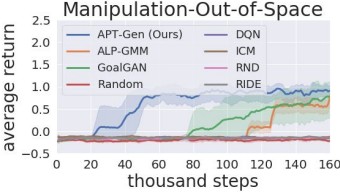

Figure 4: Results on task that is out of the predefined task space.

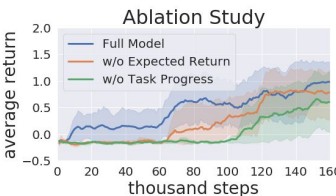

Figure 5: Results of ablation study in `Manipulation-C`.

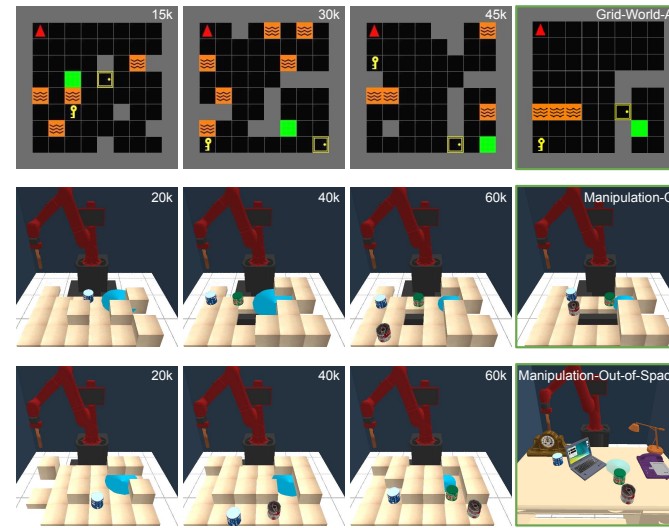

Figure 6: Progression of the generated tasks for various target tasks in the two task domains and the out-of-space task.

### 4.3 ABLATION STUDY

We conduct ablation studies on the target task of `Manipulation-C` and analyze the effect of the indicator of learning progress. As shown in Figure 5, the performance degrades when using only either the expected return or the task progress as the learning progress. The generation often adapts too fast towards the target task when only counting on the task progress, although easier tasks can still emerge during the adaptation. When generating tasks only in response to the expected return, the policy is overwhelmed by easy tasks, which retards the learning progress of the target task. In these ablations, although the knowledge gained from such tasks sometimes could still benefit the training, they do not form the ideal curricula for solving the target task. As a result, the performances achieved by these ablations are inferior to the performance of the full model of APT-Gen.

### 4.4 PROGRESSION OF GENERATED TASKS

We present qualitative results of the generated tasks in Figure 6. Each row shows three generated tasks and the target task (marked by green borderlines) with the number of collected environment steps and the task name shown on the upper right of the images. When learning for `Grid-World-A`, the task generator first creates easy tasks in which the goal (green tile) is close to the starting position of the agent (red triangle) with few obstacles in between. Between 15k and 30k steps, the task generator gradually shifts the goal to the bottom right corner as in the target task. At the same time, walls (grey tiles) are created to form rooms enclosing the goal. At around 45k steps, the door is placed on the wall to lock the room and the key is placed in a further location in the labyrinth. The agent learns to grab the key and open the door in the target task after learning to solve the generated task since the solutions now share a similar routine. In `Manipulation-C`, generated tasks start with a clear table surface and a small distance between the target object (blue can) and the goal (cyan circle). As the agent learns to tackle such easy scenarios, a green can is placed in between as an obstacle while the goal grows larger to make sure the agent can still complete the task. At 60K steps, the environment further morphs towards the target task as more obstacles being added to the scene and the goal shrinking to the correct size.

In the out-of-space task, although the more complicated table and objects cannot be generated by the procedural generation module of limited capabilities, APT-Gen gradually learns to outline the scene of the target task by utilizing the available elements such as cuboids and empty holes. By interacting with the environment and comparing experiences in both task sources, APT-Gen trains the policy to solve the out-of-space task by approximating the challenges in the target task.

## 5 CONCLUSION

To expedite reinforcement learning in hard-exploration problems, we present Adaptive Procedural Task Generation (APT-Gen) to generate suitable tasks via black-box procedural generation modules as curricula. By jointly training the task generator, the task discriminator, and the policy, APT-Gen achieves superior performances to existing exploration and curriculum learning baselines in various target tasks in the grid world and robotic manipulation domains. By adversarially training the task discriminator to estimate the similarity between the target task and generated tasks, APT-Gen demonstrates to be effective for target tasks of unknown parameterization and out of the predefined task spaces, which expands its potential use case. We hope this work could encourage more endeavors in utilizing procedural content generation for reinforcement learning.

**Acknowledgement:** We acknowledge the support of Toyota (1186781-31-UDARO) and HAI-AWS cloud credits. We would like to thank Roberto Martín-Martín, Austin Narcomey, Sriram Somasundaram, Fei Xia, and Danfei Xu for feedback on an early draft of the paper.

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

# A    ENVIRONMENT DETAILS

We describe the details of the parameterization, the state space, the action space, and the rewards of tasks from each task space. Examples of randomly generated tasks are shown in Figure 7. Most random tasks are completely infeasible or trivially easy without a learned task generator.

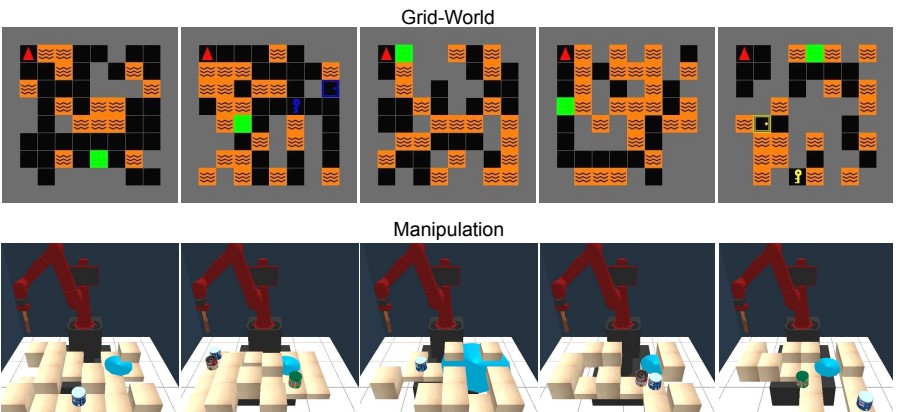

Figure 7: Examples of randomly generated tasks from the two task domains.

## A.1    GRID-WORLD.

The `Grid-World` domain is based on the popular benchmark for RL research (Chevalier-Boisvert et al., 2018). In each task, the agent (red triangle) chooses discrete actions to reach the goal (green tile) by navigating around a grid surrounded by border walls and interacting with the objects therein. Diverse labyrinths can be constructed with walls (gray tiles) and lava regions (orange tiles). When the agent hits these obstacles it receives penalties or terminates the episode. Paired keys and doors may be placed in the labyrinth for the agent to make use of. To open a door, the agent needs to first find the key of the corresponding color. The agent perceives the whole map at each step but has no prior knowledge of the functionality and rewards of the different tile types and the goals. Each episode has a finite horizon of 50 time steps. Although this task domain has a relatively shorter horizon than some of the similar hard-exploration tasks (Salimans & Chen, 2018), no sub-task reward (e.g. finding a key and opening a door) is provided as in those tasks in prior work, which introduces extra challenges to the agent.

**Parameterization.** The environment is a $10 \times 10$ grid which consists of a $8 \times 8$ configurable grid and the border walls surrounding the area. The environment is parameterized by 74 independent variables including a $8 \times 8$ array that represents the type of each tile in the grid and a 10-dimensional vector that represents the coordinates of the objects in the environment (the goal, two doors, and two keys). The x, y coordinates of the doors, the keys, and the goal in the Grid-World environment are defined as continuous variables such that they can be smoothly adapted across space. The objects are placed to the closest tile to the computed coordinates in the environment. To enable gradient descent in the task generation pipeline, the $8 \times 8$ array is converted to a $8 \times 8 \times 3$ array where the last dimension represents the logits of the tile category. The objects can be initialized to one of the $10 \times 10$ locations. If the chosen location is not on the border walls or is already occupied by a previous object, it will be placed on an empty tile there. Otherwise, the object will not appear in the environment.

**State and Action Spaces.** The agent receives the state that includes the location of the agent as a 2-dimensional vector, a local view of the surrounding tiles as a $7 \times 7$ array centered at the agent's location, and the relative positions of the objects (the goal, the doors, and the keys) as 10-dimensional vectors. If an object does not appear in the grid, the relative position will be set to $(0, 0)$. Starting at the upper left corner of the grid, the agent chooses to move along one of the four directions by one tile at each time step. If the next tile along the chosen direction is empty, the agent will be moved there. If a key is on the next tile, the agent will take the key and the corresponding door of the same color with the key will disappear.

**Rewards.** If the agent reaches the goal, the agent will succeed with a goal-reaching reward of 1. If the agent hits a wall, it will stay still and receive a penalty of 0.001. If the agent hits a lava region or a closed door, the episode will terminate with a penalty of 0.5. In addition, a time penalty of 0.001 is added to the return at each time step. If the agent is initialized on the goal, the episode will immediately terminate with a penalty of -1.

## A.2 MANIPULATION.

The `Manipulation` tasks involve a simulated robotic arm that interacts with multiple objects in a configurable table-top environment. The task is adapted from Fang et al. (2019) and simulated by a real-time physics engine (Coumans & Bai, 2016–2019). The robotic arm has the same mesh and physical parameters of a real-world Sawyer robot. The table is composed of configurable building blocks. Each block can be a flat surface, a pitfall, or a roadblock. 1 to 3 movable objects and a goal (cyan circle) are placed on the table at the beginning of each episode. The landscape of the table and the placement of the objects jointly form a puzzle in each task. The robot is asked to push the designated target object (blue can) to the goal using a tool. To achieve the goal, the robot needs to move around objects that block the way and avoid pitfalls and roadblocks. Each episode has a finite horizon of 15 steps and will terminate early if objects collide or fall off the table.

**Parameterization.** The table-top environment is formed by building blocks of a side length of 15 cm. The scene is parameterized by totally 31 independent variables including a $6 \times 4$ array that represents the types of the building blocks, a 6-dimensional vector that represents the initial position of the three objects, and a scalar that represents the range of the goal ranging from 10 cm to 50 cm. Same as in `Grid-World`, the $6 \times 4$ array is converted to a $6 \times 4 \times 3$ array where the last dimension represents the logits of the type of the building block. The objects are initially placed on the 90 cm $\times$ 60 cm table surface. Since it is in general hard for GANs and Gaussian functions to directly capture the distribution of valid positions on complex table surfaces in presence of holes and obstacles, we instead adopt a structured generation process. The model first computes a $3 \times 24$ array that represents which tile each object will be placed on. Continuous offsets ranging from -2 cm to 2 cm are then added to the x, y coordinates of the tile. If an object is initialized to a non-flat tile or the same tile with a previous object, it will not show up on the table.

**State and Action Spaces.** The agent receives the state that includes a continuous vector representing the positions of the objects and the goal and an array representing the surrounding landscape of each object. The surrounding landscape is represented by a $3 \times 4$ array where the first dimension corresponds to the object index and the second dimension corresponds to the height of the neighboring points. If the object does not appear on the table, its position will be set to $(-1, -1)$. The goal scale is not provided to the agent as part of the state and requires the agent to figure it out through trial-and-error. The robot chooses from an action space of 12 discrete actions that represent which of the three objects to push and which of the four directions to push towards.

**Rewards.** The episode will terminate with a goal-reaching reward of 1, if the target object reaches the center of the goal which is a circle of a radius of $r_1 = 10$ cm. Once the target object enters the goal region of a radius of $r_2$, it will receive a progress reward $d/(r_2 - r_1)$ where $d$ is the moving distance towards the goal. The $r_2$ is controlled by the task parameter ranging from 10 cm to 60 cm. If any object falls off the table, the episode will terminate with a penalty of -0.2. If objects collide with the roadblocks or other objects, the episode will terminate with a penalty of -0.1.

## B IMPLEMENTATION DETAILS

### B.1 NETWORK ARCHITECTURES

Neural networks are designed for the two task spaces respectively to tackle their different state spaces, action spaces, and task parameter spaces. The network architectures for each task space are shown in Figure 8 and Figure 9. The design of the task discriminator is shown in Figure 10. These neural networks are implemented with fully-connected (FC) layers, convolutional (Conv) layers, average pooling (Pool) layers, and flatten operations (Flatten). The names of different modalities of the states and the task parameters in each task domain are indicated in the figures as they are separately processed or produced. All models are implemented in Tensorflow and the hyperparameters are chosen through random search.

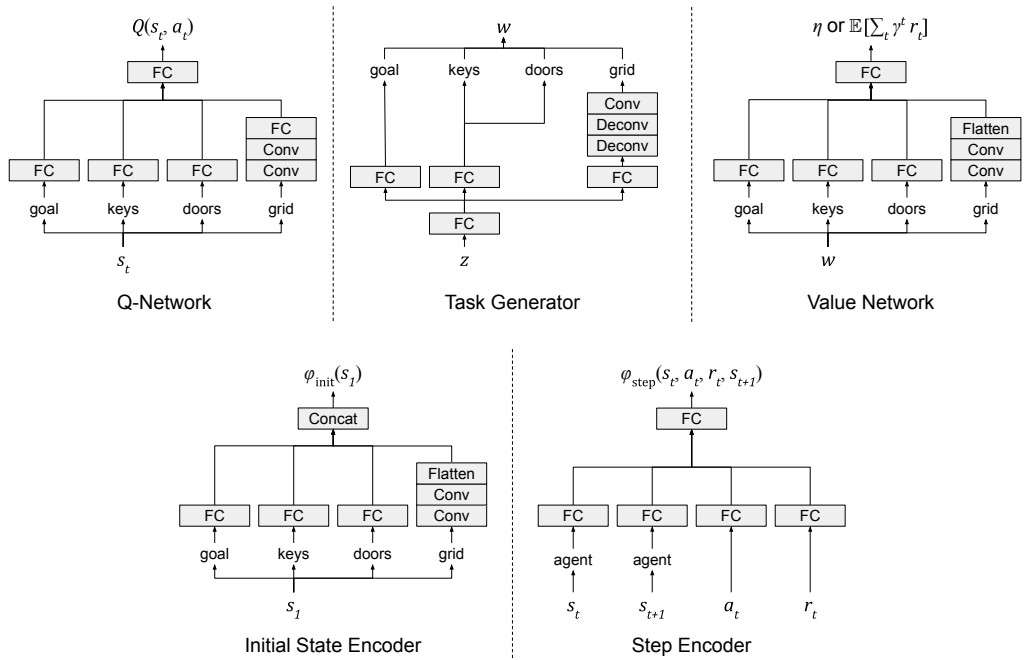

Figure 8: Network architectures for `Grid-World`.

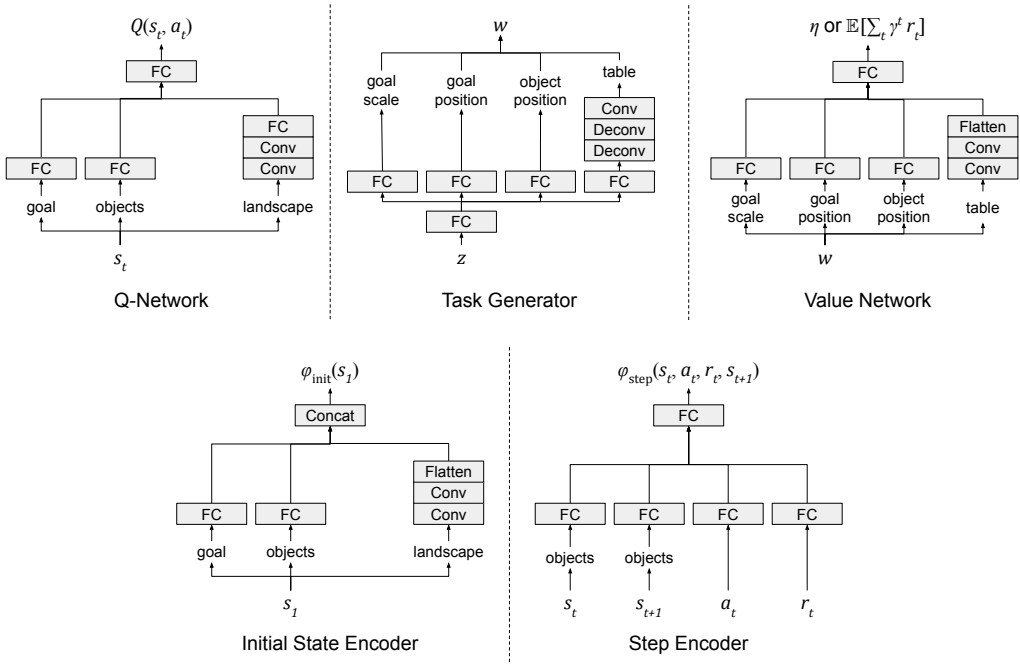

Figure 9: Network architectures for `Manipulation`.

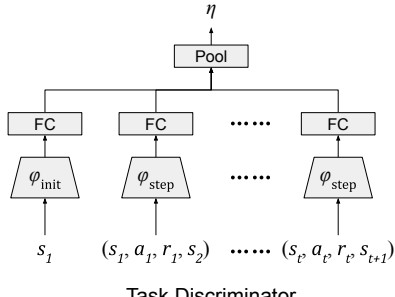

Figure 10: Network architecture of task discriminator.

**Q-Network.** The Q-network (Mnih et al., 2015) takes input as the current state and predicts the Q values for each discrete action. In both task spaces, we encode each vector in the state using a single fully-connected layer and encode the grid using two convolutional layers followed by a fully-connected layer. The information of different modalities is merged at the end with a fully-connected layer at the end. The fully-connected layers for each modality are 64-dimensional and the final layer is 128-dimensional. Each convolutional layer has a $3 \times 3$ kernel of 16 channels and a stride of 2. Each layer is followed by a rectified linear unit (ReLU). Our model and the baseline methods use the same Q-network in each task space.

**Task Generator.** The design of our task generators is inspired by generative adversarial networks (GANs) in other domains (Goodfellow et al., 2014; Radford et al., 2015). It first encodes the input noise using a 64-dimensional fully-connected layer and then separately produces each modality. The vectors are computed by fully-connected layers. We apply a sigmoid function at the output layer of each continuous modality and scale the output by the range of each modality which is defined by the task space. To compute the arrays (grid and table) of categorical values, we use a stream consists of two deconvolution layers, a convolutional layer, and a softmax layer. The convolutional layers and the deconvolution layers both have $3 \times 3$ kernels of 16 channels and a stride of 2. Following the practice of (Radford et al., 2015), batch normalization (Ioffe & Szegedy, 2015) is used in all layers of the task generators.

**Task Discriminator.** As shown in Figure 10, the task discriminator separately encodes the initial state $s_1$ using the initial state encoder $\phi_{\text{init}}$ and each transition step $(s_t, a_t, r_t, s_{t+1})$ of time step $t$ using the step encoder $\phi_{\text{step}}$. Fully-connected layers are used to predict a score for each encoding. Average pooling is applied to the predicted scores at the output layer. The architectures of $\phi_{\text{init}}$ and $\phi_{\text{step}}$ for the two task spaces are shown in Figure 8 and Figure 9. The initial state encoders are similar to the Q-networks with minor modifications to reduce the depth of the networks. The step encoder separately encodes $s_t$, , $a_t$, $r_t$, and $s_{t+1}$ and then merge the information using another fully-connected layer. In the step encoder, we do not encode the modalities in the states which do not change across time. The hyperparameters of these layers are the same as that in the Q-networks.

**Value Networks.** The value networks that are used to predict the task progress $\eta$ and the expected return $\mathbb{E}[\sum_t \gamma^t r_t]$ share the same architectures in each task space. The designs of the value networks are similar to the initial state encoders except that the inputs are the task parameter $w$ instead of $s_1$.

## B.2 ADAPTIVE ADJUSTMENT OF THE KKT MULTIPLIER

We adopt an automated procedure (Schulman et al., 2017) to adjust $\beta$ to balance the task progress and the expected return. The average expected return on the generated tasks is constantly evaluated in the replay buffer. When the average expected return exceeds a predefined threshold, $\beta$ will be scaled accordingly. Otherwise, it will remain the same. Maximum and minimum values of $\beta$ are chosen by hand to prevent the optimization to explode. Specifically, we use $\delta = 0.5$ with a tolerance of 0.1. If $\mathbb{E}[\sum_t \gamma^t r_t] < 0.4$, $\beta \leftarrow \min(\beta \times 2, 8)$; if $\mathbb{E}[\sum_t \gamma^t r_t] > 0.6$, $\beta \leftarrow \max(\beta/2, 1/8)$. In practice, we find that the performance of the policy is not sensitive to the choice of $\delta$ between 0.3 and 0.7. An ablation study on the influence of $\delta$ can be found in Sec. D.

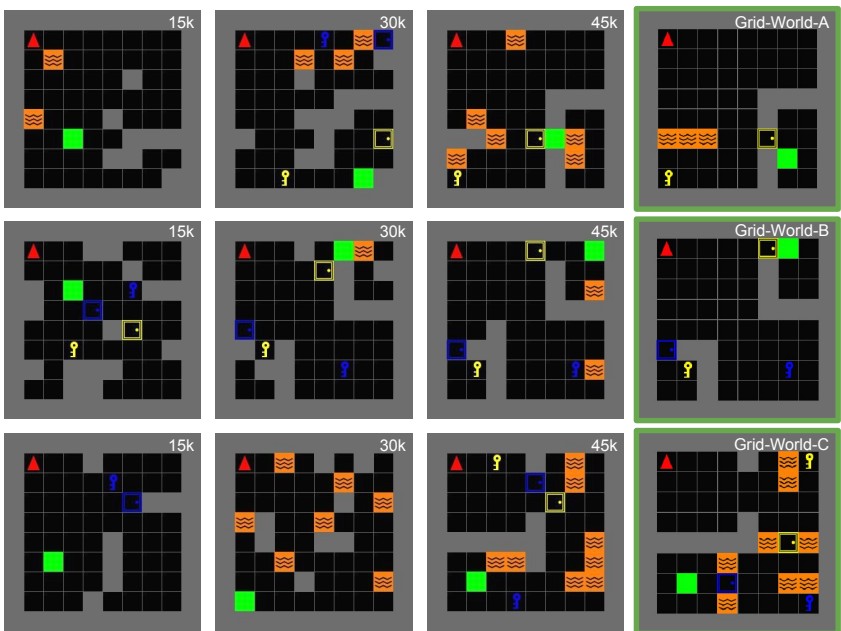

Figure 11: Progression of the generated tasks for target tasks in `Grid-World`.

## B.3 TRAINING

**Solver and Hyperparameters.** For all experiments, we use the ADAM optimizer (Kingma & Ba, 2014) with learning rate of $3 \times 10^{-4}$, $\beta_1 = 0.9$, $\beta_2 = 0.999$ and the batch size of 128. Totally 10,000 environment steps are collected to initialize the replay buffers. We only sample from the most recent 10,000 steps in each replay buffer to train the task discriminator to encourage it to estimate the return of the updated policy. After collecting each environment step, the policy is trained for 10 iterations and the other models are trained for 1 iteration. The $\beta$ is updated every 500 environment steps according to the average returns of the most recent 50 episodes. All hyperparameters are chosen by random search.

**Stability.** One of the technical challenges is that the training could be unstable when the policy learns to solve a changing distribution of tasks, which is a common issue in related problems such as continual learning (Kirkpatrick et al., 2017). This issue is intensified in the Grid World task space since the reward is more sparse. While fundamentally resolving this issue of reinforcement learning in non-stationary task distribution is beyond the focus of our focus in this paper, we propose two simple techniques to improve the stability of the RL agent during training. First, as the stability is often affected by overestimation of the Q-values, we clip the next Q-values in the DQN algorithms by the maximum and minimum episodic return observed in the replay buffer. Second, we restore the policy network's parameters if its performance in the target task has reached above 0 but drops since the last round of evaluation. We found that these two modifications to the original DQN algorithm significantly improves the stability of the training in APT-Gen and baseline methods.

**Computation and Runtime.** During each run, the method is trained on a single NVIDIA GeForce GTX1080 Ti GPU and 8 CPU cores with 32 GB memory. The overall data collection and training time of each run takes around 2 hours for Grid World and 30 hours for Robotic Manipulation.

## B.4 EVALUATION

The evaluation is conducted after collecting every 1,000 environment steps. In all quantitative evaluation in this work, each data point is evaluated for 50 episodes. The means and the error bars of the episode returns are computed across 5 different runs of training the same method. To have fair comparisons, all methods are trained with a fixed budget of total steps collected by the agent.

## C  ADDITIONAL RESULTS ON TASK PROGRESSION

In Figure 11 and Figure 12, we present more examples of the generated tasks for the two domains. Each row shows three generated tasks and the target task (marked by green borderlines) with the number of collected environment steps and the task name shown on the upper right of the images. As shown in the figure, the tasks generated by APT-Gen form curricula of ascending difficulties and similarity with the target task during training.

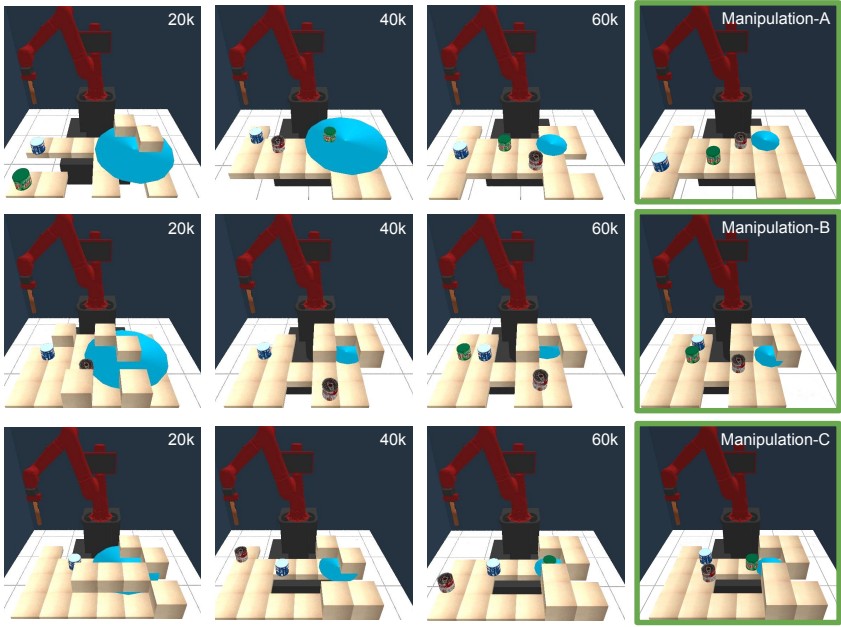

Figure 12: Progression of the generated tasks for target tasks in `Manipulation`.

## D  ABLATION ON MINIMUM EXPECTED RETURN

In this section, we conduct an additional ablative experiment to demonstrate the influence of the minimum expected return $\delta$. We train APT-Gen in `Manipulation-C` with different choices of $\delta$ ranging from -0.1 to 1.1 for 5 runs each. The training curves are shown in Figure 13 where we report the average returns achieved by the policy and the shaded area indicates the standard deviation across different runs. Using $0.3 \leq \delta \leq 0.7$, the learned policies can achieve similar performances. Using $\delta$ outside of this range, the feasibility of the generated tasks becomes less distinguishable by the threshold and the performance of the policy decreases.

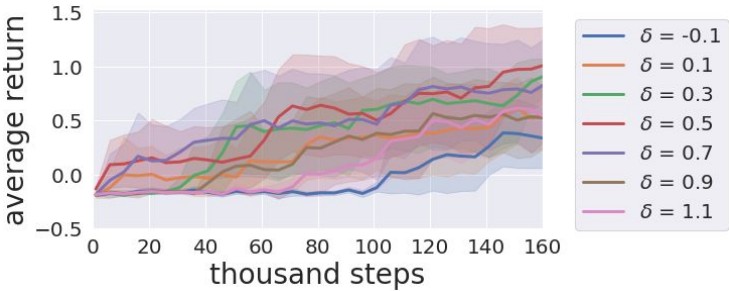

Figure 13: Ablation on the minimum expected return.

# E    DISCUSSION AND FUTURE WORK

While APT-Gen is able to procedurally generate tasks as curricula for various target tasks, it does have limitations. These limitations will lead to several exciting future directions. First, without a structured representation of the task, it is often hard for the task generator to produce task parameters that will result in useful and feasible tasks. Since our task generator directly uses convolutional layers to produce the category of each tile, it usually takes a large number of steps before it learns to properly form the walls of rooms and place the doors in the right place. This can potentially be improved with recent advances of deep generative models using geometric structures (Tian et al., 2019; Huang et al., 2020a). Second, this work focuses on learning to generate tasks as curricula for a single target task. Extending APT-Gen to generate tasks for multi-task learning and meta-learning would be a promising future direction. Lastly, while the parameterized task spaces in this work contain rich variations of the environment and the objects therein, the structure and properties of the robotic agent are fixed. It would be interesting to combine APT-Gen with recent works on morphology (Ha, 2018; Huang et al., 2020b) to adapt the agent in the task.

