# OpenReview forum: "Adaptive Procedural Task Generation for Hard-Exploration Problems"
_ICLR.cc/2021/Conference — ICLR 2021 Poster_

### Official Review · AnonReviewer1 · 2020-10-27
**This paper proposes a training framework to faciliate reinforcement learning in hard-exploration problems. The authors adopted ideas in curriculum learning and used a task generator to adaptively generate tasks that faciliate RL agent's learning. An adversarial learning frame work was used to train both the task generator and a discriminator which estimates differences between generated tasks and real target tasks. Their method show superior performance in several RL training scenarios.**

**Rating:** 6
**Confidence:** 4

**Review:**


This paper tackles the problem of facilitating RL agents' learning in sparse reward, hard-exploration problems. The authors approached this challenge by generating a curriculum of tasks needed to finish the originally assigned task. Though using other auxiliary tasks to assist RL training has been heatedly discussed, their method has its own highlights and novelty. Pros and Cons are listed as follows:

[+] The overall framework is illustrated with clear explanations. The formulation of adversarial learning over the unrolling trajectories $\tau$ is the key for connecting the task generator and discriminator in this case and is interesting. The implicit formulation of task generation provided the prerequisite for solving out-of-scope tasks (when only $\mathcal{S}$ and $\mathcal{A}$ are shared).

[+] The experiments are well-designed with illustrative figures and analyses. Through the analysis, we see the advantage of having such curriculums do benefit agents' learning compared to previous methods. The ablative study also gave an intuitive explanation of the balance between having more difficult tasks in the curriculum and agents' performance on the original task.

[-] Although the general idea of generating easier tasks to facilitate agents' learning in complex tasks is intuitive, how the authors  justify their method has the same property is not fully justified in my mind. As far as I'm understanding, the authors used a hyperparameter $\delta$ to avoid making task generator $G$ moves too fast towards the target task, however, as was mentioned in Appendix B.2, the selection of the KKT multiplier is not critical, which causes some confusion on how the step-by-step difficulty enhancement was achieved. This result discussed is also having me question what if the task and reward scale is completely different, then would the effect of alternating $\beta$ still be small?

[-] The second problem is perhaps with the starting states in this training process. As the generator and discriminator was trained to adaptively fit the current RL agent's performance, the initiation of task generation should be important as a completely different setting of environment even with same state space and action space should be harmful for RL training in my mind(since there might be conflicts in $D_g$ and $D_{target}$ ). Then in this case, for example in a simpler scenario,  will this be a problem that affects RL training? Do we need to add constraints to the task generator at the beginning? This further leads to the question of whether the performance boost was provided by sampling on two similar environments (or even same as $G$ moves fast to the assigned task) and leading to more trajectories for RL training, or was the performance boost provided by the curriculum mentioned.

Because of the aforementioned questions, a borderline decision was made for now, please check the questions in the [-] to clear any misunderstandings.

=====================================================================================================

The authors addressed my concerns in the discussion period and I therefore created my score to 6.

---

> ### Author Response · Authors · 2020-11-16
> **Our Response to Reviewer 1**
>
>
>
> Thank you very much for your informative and constructive comments.
>
> 1\. Choice of the hyperparameter $\delta$
>
> We’d like to clarify that the suitable range of $\delta$ indeed depends on the task and the scale of the reward, whereas a wide range of $\delta$ can result in similar performances. We have conducted an additional ablative experiment to fully demonstrate the influence of $\delta$. We train APT-Gen in Manipulation-C with different choices of $\delta$ ranging from -0.1 to 1.1 for 5 runs each. The training curves are shown in Figure 13 in the updated version of the paper. We also summarize the mean and standard deviation of the return at the end of the training as below:
>
> | $\delta$  | return |
> |---|---|
> |-0.1 | 0.521 $\pm$ 0.366 |
> |0.1 |  0.769 $\pm$ 0.108|
> |0.3|  0.943 $\pm$ 0.207|
> |0.5| 1.026 $\pm$ 0.265|
> |0.7| 0.894 $\pm$ 0.281|
> |0.9| 0.797 $\pm$ 0.140|
> |1.1| 0.728 $\pm$ 0.106|
>
> In the paper, we train the model with $\delta = 0.5$, which archives an average return of 1.026 after training. Using $0.3 \leq \delta \leq 0.7$, the learned policies can achieve comparable performances. Using $\delta$ outside of this range, the feasibility of the generated tasks become less distinguishable by the threshold and the average return decreases. We appreciate the suggestion and have adjusted our description in the updated paper to “In practice, we find that the performance of the policy is not sensitive to the choice of $\delta$ between 0.3 and 0.7.” to avoid confusion in the future. This additional experiment has been added to Sec. D in the updated version of the paper.
>
> 2\. Initialization of the task generator
>
> While we agree that a carefully hand-designed initialization could enable APT-Gen to work better in complex task spaces, in this paper we focus on the general case where the task generator is randomly initialized. In practice, this has been working effectively in our experiments. At the beginning of training, the task generator will first generate easy tasks (e.g. reaching a goal that is close to the agent). Although these tasks have different environment layouts and goal positions, they often share similar challenges faced by the target task such as reaching the goal and avoiding obstacles. Therefore, the knowledge gained in these easy tasks could still be useful for learning to solve the target task. As demonstrated in Sec. 4.4, we find the generated tasks will become similar to the target task after a few thousand iterations.
>
> The performance boost has nothing to do with the number of trajectories in our experiments, since all methods are trained with **the same number of collected steps** as explained in Sec. 4.2. To have fair comparisons, the x-axes of APT-Gen and curriculum learning baselines (Random, GoalGAN, ALP-GMM) indicate the total steps collected from both target and generated tasks, while x-axes of other baselines (DQN, ICM, RND, RIDE) indicate the steps collected from only the target task. Therefore, APT-Gen will not receive more trajectories than other methods.
>
> We hope these additional explanations and experiments have addressed your previous questions. Please don’t hesitate to let us know if any further clarification would help you reconsider your rating.

---

> > ### Comment · AnonReviewer1 · 2020-11-24
> > **Response to the author**
> >
> > Thank you for your response. The ablative study on $\delta$ clear my confusion, I assume we will still need to choose a rather good $\delta$ in other experiments but there will also be a similar range where the advantage of the current method shows up. I appreciate the authors' effort in answering my questions and solving them with extra qualitative analysis if needed.

---

> > > ### Author Response · Authors · 2020-11-24
> > > **Thanks for your comments**
> > >
> > > Dear Reviewer 1,
> > >
> > > It is great to hear that your questions have all been cleared. We’d like to thank you again for your constructive comments, which helped us improve the quality of the paper.

---

### Official Review · AnonReviewer3 · 2020-10-29
**This paper shows some promising results, but the claims concerning the existing algorithm in the literature should be more clearly supported and demonstrated in the experimental section.**

**Rating:** 4
**Confidence:** 4

**Review:**

In the paper "Adaptive Procedural Task Generation for Hard-Exploration Problems", the authors introduce a new method to automatically generate a curriculum for reinforcement learning algorithms for hard-exploration problems.
The approach uses a generative network to generate variants of the agent's environment with varying difficulty and then a discriminator to automatically access the resemblance of the generated environment with the target environment. The generating process then uses this resemblance metric and the observed task return to progressively drive the learning process toward environments that can be solved by the agent and that become closer and closer to the target environment.

The paper is well structure and clearly written. The illustrations, in particular, Figure 1 are particularly helping to understand the overall process of the algorithm.


There are, however, a few elements that can be improved in the paper.

First, in several places, the paper is quite vague on the claim and differences with other algorithms. For instance, "In contrast, we propose a general framework for highly configurable task spaces with parameters of much higher dimensions.". What is the exact scale of the difference? I think the paper will be more convincing with precise examples and the exact numbers of the dimensions.

Related to this aspect, the task description provided in the paper is not sufficient to fully understand what are the degrees of freedom given to the generative network to generate different environments. In particular, what is changing between the different instances? This aspect is too central for the paper that it cannot be relegated to the appendix.

After reading the appendix, we can then find the parameters of the environment are all discrete, with few possible values. This is a big deal actually because even if the number of parameters is larger in this paper, other papers referred here are using continuous parameter space, which is by definition infinitely larger. Thus the claims made in the paper need to be tuned down a little bit to underline this difference, and it could be important to discuss how the proposed approach to scale to continuous parameter spaces.

Similarly, the justification of the baseline selection is not sufficient in the paper to understand the motivations of the authors and why these algorithms are used in this study, and also why some approaches are not included. For instance, the POET seems to follow a similar approach, yet working on continuous parameter spaces. According to the authors, the propose methods will work better because it is not trying to cover all the space of possible environment, but this remains to be demonstrated in the experimental section.

Overall the method proposed in this paper shows some promising results, but the claims concerning the existing algorithm in the literature should be more clearly supported and demonstrated in the experimental section.

---

> ### Author Response · Authors · 2020-11-16
> **Our Response to Reviewer 3  (Part 2)**
>
> 4\. Justification of the baseline selection
>
> The 7 baselines are carefully chosen to cover state-of-the-art approaches along different lines of research, including 3 curriculum learning methods (Random, GoalGAN, ALP-GMM), 3 exploration strategies (ICM, RND, RIDE), and a model-free RL baseline. To conduct thorough comparisons, we evaluate all **8 methods** in **7 target tasks** from **2 different task domains**, including an out-of-space manipulation task. To perform in-depth analysis, we further provide detailed qualitative results and ablation studies in the paper. APT-Gen consistently outperforms all baselines including the three curriculum learning methods that indiscriminately solve all tasks within the task space, which we believe is a convincing proof of the advantage of our approach.
>
> While we agree that POET is an exciting related work and we have acknowledged its significance with full respect, it is proposed for a different problem formulation. As indicated by its name, POET aims to discover diverse task environments in an open-ended manner and trains a different policy for each paired task. In contrast, APT-Gen is a target-oriented approach to progressively generate a sequence of tasks as curricula for learning a policy to solve the target task. Without the notion of target task, POET does not encourage the generated tasks to be useful or similar to the problem of interest. Even if POET occasionally discovers tasks that are similar to the target task by chance, it lacks a mechanism to identify the suitable policy for solving the target task from the large number of policies it produces. Therefore, although POET shares the similar spirit of generating tasks from a parameterized task space, it is not directly comparable to our approach or other baselines evaluated in this paper.
>
> We sincerely believe that a significant amount of efforts have been made in this paper to support our claims, through a variety of baselines, the diverse set of target tasks, and the detailed analysis. In the updated version of the paper, we have polished and adjusted our descriptions. We hope these additional explanations and improvements have addressed your previous questions. Please let us know if any further clarification or adjustments would help you reconsider your rating.
>
> [1] Portelas et al. “Teacher Algorithms for Curriculum Learning of Deep RL in Continuously Parameterized Environments.” In CoRL, 2019.
>
> [2] Wang et al. “Paired Open-Ended Trailblazer (POET): Endlessly Generating Increasingly Complex and Diverse Learning Environments and Their Solutions.” In arXiv, 2019.
>
> [3] Matiisen et al. “Teacher-Student Curriculum Learning.” In arXiv, 2017.
>
> [4] Florensa et al. “Automatic Goal Generation for Reinforcement Learning Agents.”. In ICML 2018.

---

> > ### Comment · AnonReviewer3 · 2020-11-23
> > **Additional question**
> >
> > Dear authors,
> >
> > Thank you for this detailed answer.
> >
> > Could you please clarify the following?
> > In your response, you say: "The x, y coordinates of the doors, the keys, and the goal in the Grid-World environment are defined as continuous variables such that they can be smoothly adapted across space. The objects are placed to the closest tile to the computed coordinates in the environment. "
> >
> > The way I understand the last sentence is that these parameters are eventually discretised, even if they come from a continuous parameter. Therefore, out of the 31 and 74 variables considered in the paper, only the goal location in the manipulation task is continuous.
> >
> > Am I correct?

---

> > > ### Author Response · Authors · 2020-11-23
> > > **Thanks for your comments. Further clarification.**
> > >
> > > Dear Reviewer 3,
> > >
> > > Thank you again for your comments.
> > >
> > > In the Manipulation tasks, we’d like to clarify that it is the range of the goal instead of the goal location that is continuous. In the Grid World tasks, the x, y coordinates are computed from the continuous parameter space such that they can be smoothly adapted across space, and then the objects are indeed placed on discretized tiles in the grid eventually. The same grid world task domain has been widely used for evaluation in prior work on RL and hard-exploration problems, including the RIDE baseline compared in our experiments. In the Manipulation tasks, however, the x, y coordinates of objects are continuous variables ranging from 0 cm to 60 cm and from -45 cm to 45 cm in the simulated environment. Since it is in general hard for GANs and Gaussian functions to directly capture the distribution of valid positions on complex table surfaces in presence of holes and obstacles, we instead adopt a structured generation process. In our model as well as other curriculum learning baselines, it first computes which table tile each object should be placed on, and then continuous offsets are added to the x, y coordinates of the tile making them continuous values eventually. We have included detailed descriptions of these implementation details in the updated paper.
> > >
> > > In the updated paper, we have followed your suggestions and revised Sec. 2 to provide more precise descriptions to better acknowledge the prior work. If you would like to kindly provide us any additional suggestions on how to make the descriptions related to this or any other of your previous questions even more rigorous, we would be happy to make further updates on the paper.

---

> ### Author Response · Authors · 2020-11-16
> **Our Response to Reviewer 3 (Part 1)**
>
> Thank you very much for your informative and constructive comments.
>
> 1\. Exact number of task parameters and exact scale of difference
>
> Both of these two numbers have been described in detail in the main paper:
> -   The exact number of task parameters can be found at the beginning of **Sec. 4.1 Tasks**: "The Grid-World tasks are parameterized by $74$ variables and the Manipulation tasks are parameterized by $31$ variables."
> -   The exact scale of difference can be found in the last paragraph of the **Sec. 1 Introduction**: "Tasks generated in these domains are parameterized by $6 \times$ to $10 \times$ independent variables compared to those in prior work [1, 2]."
>
> In the updated paper, we highlight these numbers and replace the phrase "much higher dimensions" with more detailed explanations in Sec. 2 to avoid confusion.
>
> 2\. What is changing between the different task instances?
>
> The variations of task instances and degrees of freedom are described in these sections and figures:
> -   In **Sec. 4.1**, we provide high-level descriptions of the variation of environment designs and constraints (e.g. lava regions, object positions).
> -   In **Sec. 4.4**, we elaborate on the variations of generated tasks across the training time.
> -   In **Sec. A Environment Details**, all details of the task design, parameterization, state and action spaces, and reward functions are provided.
> -   In **Figure 2**, various examples of target tasks in the two task domains are illustrated.
> -   In **Figure 7**, we further provide 10 examples of randomly generated tasks from the two task spaces.
>
> We agree that some details can be mentioned earlier in the paper. Therefore, we have added additional explanations in Sec. 1 and Sec. 4.1: “Each task instance can have different environment layouts, object types, object positions, constraints, and reward functions.” and “Each task space contains various tasks that share the same state and action spaces but different environment layouts, object types, object positions, constraints, and reward functions.” Please see the updated paper for details.
>
> 3\. Discrete and continuous task parameters
>
> First, we’d like to clarify that not all of our task parameters are discrete and our task parameter spaces contain not a few but a large number of possible values. As described in Sec. A, the range of the goal in the Manipulation tasks is a continuous variable ranging from 10 to 50 cm. The x, y coordinates of the doors, the keys, and the goal in the Grid-World environment are defined as continuous variables such that they can be smoothly adapted across space. The objects are placed to the closest tile to the computed coordinates in the environment. The $8 \times 8$ and $6 \times 4$ grids and 3 different tile categories result in $3^{64}$ and $3^{24}$ possible values, which are huge spaces to sample from. Finding the suitable task parameters in such large parameter spaces is a nontrivial problem and is significantly harder than choosing from a small set of predefined tasks as in prior work [3]. We have included additional explanations in Sec. 4.1. and Sec. A to make these details clearer.
>
> Second, we emphasize that our approach does not rely on any assumptions about the task parameters and has demonstrated superior performance to baselines originally proposed for either continuous [1, 2, 4] or discrete [3] task parameters. The task discriminator of APT-Gen provides a smooth signal to adapt the task generator, which enables it to efficiently work with a mix of continuous and discrete task parameters. As shown in our experiments in Sec. 4.2, APT-Gen successfully learns to solve various target tasks while the state-of-the-art curriculum learning baselines fail in many challenging tasks.
>
> Lastly, it is actually not always harder to generate continuous parameters than discrete parameters, although we agree that adding more continuous parameters can make the tasks even more interesting. While continuous spaces contain infinite possible values, combinatorial search in discrete spaces can often be more challenging to solve. Moreover, we find that what usually matters significantly to task generation is whether a small change in the task parameter space will alter the desired solution to the task. For instance, whether the depth of a gap is 1 meter or 1.5 meters might not affect how the agent steps across it, but whether a tile on the critical path is empty or blocked can completely change the way to reach the goal.
>
> In the updated paper, we have highlighted the differences of task parameterization to better acknowledge prior work and have adjusted the description to: “Our approach is able to effectively generate tasks from task spaces parameterized by 74 and 31 variables. The task parameters consist of a combination of high-dimensional discrete variables and several continuous variables.” Please don't hesitate to let us know if you suggest any further adjustments that will make this part more rigorous.

---

### Official Review · AnonReviewer2 · 2020-10-29
**Official Blind Review #2**

**Rating:** 7
**Confidence:** 3

**Review:**

This paper presents a procedurally content generation approach (APT-Gen) that generates a sequence of tasks for an agent to solve. These tasks are automatically generated in a way that helps an RL agent to learn hard-exploration problems. A main innovation of the approach is a task generator system that is both rewarded for generating tasks the agents can solve but also a sequence of tasks that are getting increasingly more similar to the target task.

I find the paper interesting, the results convincing and don't have too much to critique. My main critique would be to add what the limits of the current approach are. What are the hard-exploration problems it fails on and how much can the target task be outside the predefined task space before performance degrades? A more systematic study of these limitations would further add to the paper and highlight potential future directions for improvements.

Minor comments:
- Based on how many independent runs are the results based on that are shown in Figure 3, 4, 5?  What do the shaded areas show? One standard deviation?
- A related paper that does adaptively and procedurally generate levels, instead of tasks, is "Illuminating Generalization in Deep Reinforcement Learning through Procedural Level Generation " Justesen et al., 2018

---

> ### Author Response · Authors · 2020-11-16
> **Our Response to Reviewer 2**
>
>
> Thank you very much for your encouraging and constructive comments.
>
> 1\. Evaluation details
>
> Each method is evaluated for 5 independent runs in these results as explained in Sec. B.4. The solid curve indicates the means of the reward and the shaded area indicates the standard deviation across different runs. To make this clearer in Sec. 4.2, we have added: “The average return across 5 independent runs are reported and the shaded area indicates the standard deviation.”
>
>
> 2\. Additional related work
>
> Thank you for suggesting this related work. We found this paper very relevant and have updated our paper to include it in the related work section.
>
>
> 3\. Discussion and future work
>
> We appreciate your suggestion and agree that such discussions would make our paper stronger. We have included an additional section of Discussion and Future Work in the updated version of our paper:
>
> While APT-Gen is able to procedurally generate tasks as curricula for various target tasks, it does have limitations. These limitations will lead to several exciting future directions. First, without a structured representation of the task, it is often hard for the task generator to produce task parameters that will result in useful and feasible tasks. Since our task generator directly uses convolutional layers to produce the category of each tile, it usually takes a large number of steps before it learns to properly form the walls of rooms and place the doors at the right place. This can potentially be improved with recent advances in deep generative models using geometric structures [1, 2]. Second, this work focuses on learning to generate tasks as curricula for a single target task. Extending APT-Gen to generate tasks for multi-task learning and meta-learning would be a promising future direction. Lastly, while the parameterized task spaces in this work contain rich variations of the environment and the objects therein, the structure and properties of the robotic agent are fixed. It would be interesting to combine APT-Gen with recent works on morphology [3, 4] to adapt the agent in the task.
>
> We have also listed all other improvements in our general response above. Please don’t hesitate to let us know for any additional comments on the paper or on the changes.
>
>
> [1] Tian et al. ”Learning to Infer and Execute 3D Shape Programs.” In ICLR, 2019.
>
> [2] Huang et al. “Generative 3D Part Assembly via Dynamic Graph Learning.” In NeurIPS, 2020.
>
> [3] David Ha. “Reinforcement Learning for Improving Agent Design”. In arXiv, 2018.
>
> [4] Huang et al. “One Policy to Control Them All: Shared Modular Policies for Agent-Agnostic Control. “ In ICML, 2020.

---

> > ### Comment · AnonReviewer2 · 2020-11-23
> > **Follow-up**
> >
> > Thank you for the response. The added discussion on limitations of the approach is very useful and appreciated.

---

> > > ### Author Response · Authors · 2020-11-24
> > > **Thanks for your feedback**
> > >
> > > Dear Reviewer 2,
> > >
> > > Thank you again for your encouraging and constructive feedback, which helped us improve the quality of the paper.

---

### Official Review · AnonReviewer4 · 2020-11-01
**Review for paper "Adaptive Procedural Task Generation for Hard-Exploration Problems"**

**Rating:** 6
**Confidence:** 4

**Review:**

Summary:
This paper proposes a framework to progressively generate a sequence of tasks as curricula for hard-exploration tasks. It demonstrates better performance on MiniGrid and robot manipulation tasks compared with exploration baselines and curriculum goal generation baselines.

Pros
+ The paper is well-written and easy to follow. It provides enough implementation details.
+ This paper tackles a hard problem that incorporates exploration, policy learning, and curriculum task generation. The adversarial training solution is clean. It proposes a new way to learn the similarity score by encoding the MDP.

Questions & Concerns:
- The optimization problem on page 4.
 1. it's not an equation, it's an optimization problem. There are many references in the paper using "Eq. 1", "Eq. 2" for the optimization problem.
 2. There is nothing related to KKT condition for (2). KKT condition is some first-order necessary conditions for the saddle-point to be optimal. Form (2) in the paper doesn't use KKT. It's just a Lagrangian form.
 3. $\beta$ is usually called the Lagrangian multiplier rather than the KKT multiplier.
 4. Since $\beta$ is the dual variable, the optimization problem for $\beta$ is $min_{\beta} \text{ }L$ with $\beta$ lower bounded by *zero*. So that this problem is the upper bound of the primal problem. It shouldn't *explode* when you optimizing $\beta$ since you can just choose $\beta$ to be small enough to get close to the primal problem.
 5. I understand the author may use some empirical way (an optimization problem with regularization) as they showed in their appendix, rather than primal/dual optimization. It's better to make this part rigorous.
- Can you explain why $V_1$ and $V_2$ only take $w$ as input? Both $V_1$ and $V_2$ are some total expected returns (of rewards or similarity), so they are also related to the policy, not only the tasks. In the original deep RL framework, a value network should take states as input. In your setting, it should take the states, or trajectory (embedding) as input. It's not reasonable to have a value estimation that doesn't take the data samples as input. Do you assume $V_1$ and $V_2$ can always fit the returns generated by an arbitrary policy?
- In the algorithm, do you have any inner loops to train the discriminator and the RL algorithm? By meaning "while not converge", do you mean the convergence for all the losses as well as the RL performance? Will they work in a coherent way?
- In Fig 5, it seems APT-Gen without task progress can also work to some extent. This is confusing. Why the policy is not stuck on easy scenarios?
- (Minor concern) POET is an important baseline but the results are not included.

I'm willing to change my score if some concerns are addressed.

========post rebuttal review=========

After reading the response from the authors and the new version of this paper, I decided to increase my score to 6.

---

> ### Author Response · Authors · 2020-11-16
> **Our Response to Reviewer 4**
>
>
> Thank you very much for your informative and constructive comments.
>
> 1\. Notations of the optimization problem
>
> We sincerely appreciate the detailed suggestions about the notations of the optimization problem. We initially adopted the notations from prior work [1]. After carefully considering these comments, we agree that these suggestions would make the paper more rigorous. We have revised the paper with these suggested notations. Please see the updated paper for details.
>
> 2\. Inputs of the value functions $V_1$ and $V_2$
>
> Instead of fitting values for arbitrary policies, $V_1$ and $V_2$ aim to fit the expected values $\mathbb{E}_{\tau \sim M(w), \pi} [ \cdot ]$ which are defined over the distribution of the trajectory $\tau$ given $w$ and the current $\pi$ as explained in Sec. 3.2. To this end, the training data used for fitting $V_1$ and $V_2$ are collected using $\pi$ instead of arbitrary policies. Therefore, $\tau$ is not provided as part of the inputs to $V_1$ and $V_2$.
>
> Although we agree that a more general definition of $V_1$ and $V_2$ could include the model parameters of $\pi$ as the "state" of the teacher-student paradigm, it is not necessary for the teacher-student curriculum learning formulation in this paper. Actually, this has been discussed by the original Teacher-Student Curriculum Learning paper [2] and they also omit the policy parameters in their value function. Similarly, we focus on the curriculum learning problem in this paper, where the eventual goal is to solve the target task by generating a sequence of tasks as curricula. Unlike the RL analogy where fitting the value of a previously observed state $s$ can help the policy to choose action when similar states are re-visited in future episodes, $G$ and $\pi$ keep adapting towards the target task and usually the previous policy parameters will not be re-visited. Thus feeding the policy parameters to $V_1$ and $V_2$ will be unnecessary and distractive. Therefore, we follow the practice of prior work [2] and do not provide the policy parameter to the value functions as inputs.
>
> Feeding in the policy parameters could be more reasonable for task generation in multi-task learning and meta-learning problems, where the learned value functions can be reused to generate curricula in multiple runs for different target tasks. That could be an exciting future direction of this work.
>
> 3\. Details of the training loop
>
> The training of the task generator, the task discriminator, and the policy are conducted in the same loop. The training loop terminates when all the losses converge, including the loss for reinforcement learning. As shown in our quantitative results, the training pipeline works coherently and the agent can successfully learn to solve a variety of target tasks. We have updated Sec. 3.3 to make this clearer in the paper.
>
> 4\. Ablation without task progress
>
> Without the task progress, the task generator will focus on generating easy tasks at the beginning. Although these tasks can be very different from the target task, they could share similar challenges faced in the target task, such as reaching the goal and avoiding obstacles. As described in Sec. 3.1 and 3.3, the policy is trained on rollouts collected from both the generated tasks and the target task. In this way, easy tasks can still serve as useful auxiliary tasks for training the policy, although they do not form an ideal curriculum for solving the target task. As a result, the performance achieved by this ablation is much worse than the full model of APT-Gen but better than the model-free RL baseline trained only on the target task.
>
> 5\. Comparison with POET
>
> We’d like to clarify that POET and APT-Gen are proposed for different problems, although the two approaches share a similar spirit of generating tasks from parameterized task spaces. POET, as indicated by its name, aims to discover diverse task environments in an open-ended manner and trains a different policy for each paired task. Without the notion of target task, POET does not encourage the generated tasks to be useful or similar to the problem of interest. Even if POET occasionally discovers tasks that are similar to the target task by chance, it lacks a mechanism to identify the suitable policy for solving the target task from the large number of policies it produces. Therefore, POET is not directly comparable to our approach and other baselines evaluated in the paper.
>
> We have also listed all other improvements in our general response above. Please don’t hesitate to let us know for any additional comments on the paper or on the changes.
>
> [1] Higgins et al. “beta-VAE: Learning Basic Visual Concepts with a Constrained Variational Framework.” In ICLR, 2017.
>
> [2] Matiisen et al. “Teacher-Student Curriculum Learning.” In arXiv, 2017.

---

> > ### Comment · AnonReviewer4 · 2020-11-24
> > **Response to the author**
> >
> > Thank you very much for your feedback.
> >
> > For 1, 3, 4, 5, I'm convinced by your response. And I'm very happy to see you made revisions to make this work more complete and rigorous.
> >
> > For 2, what I mean by "arbitrary policy" is because $V_1$ and $V_2$ doesn't use any policy information as input. So no matter what $\pi$ is, these $V_1$ and $V_2$ will always be able to fit the expected values $\mathbb{E}_{\tau \sim M(w), \pi} []$ given $w$ as input. For the single-task analogy (the original RL setting), it sounds like learning a value function only from a fixed (random) task vector.
> >
> > I understand providing policy parameters to $V_1$ and $V_2$ will cause a lot of problems for the training. However, instead of using the parameters of a policy distribution, we can use samples to represent a policy, which is the trajectory $\tau$. So when taking a summation in $\tau$ ($\mathbb{E}_{\tau \sim M(w), \pi} []$), it's very reasonable to make $V_1$ and $V_2$ take states as input, which is a natural analogy to original deep RL framework.
> >
> > Moreover, the analogy to original RL is also reasonable, since the value (both reward-based and similarity-based) for $s$ can indeed be reused.

---

> > > ### Author Response · Authors · 2020-11-24
> > > **Thanks for your comments**
> > >
> > > Dear Reviewer 4,
> > >
> > > Thank you very much for letting us know that most of your concerns have been convincingly addressed. We are very glad to hear that you are satisfied with our revision of the paper.
> > >
> > > For the value functions, we'd like to further clarify that $V_1$ and $V_2$ will **not** fit arbitrary expected values regardless of the learned policy $\pi$. Although not explicitly input to the value functions, the policy information is incorporated into $V_1$ and $V_2$ through the training data of trajecotries $\tau$ collected by the learned policy $\pi$. As indicated by the definition of the two expectation terms, the distribution of $\tau$ is jointly determined by the task parameter $w$ and the learned policy $\pi$, instead of arbitrary policies. As explained in Sec. 3, $V_1$ and $V_2$ are jointly trained with $\pi$ and continuously adapt with respect to the updated $\pi$. To this end, $V_1$ and $V_2$ are trained using the trajectory data collected by the learned policy $\pi$. In our implementation (as detailed in Sec. 3 and Sec. B.3), we use replay buffers of limited sizes to store the collected trajectories such that only the recent trajectories collected by the updated $\pi$ will be kept in the replay buffers for training $V_1$ and $V_2$. Therefore, the value functions are not learned from a fixed/random task parameter or policy. Instead, $V_1$ and $V_2$ always aim to fit the expected values $\mathbb{E}_{\tau \sim M(w), \pi} [ \cdot ]$ with respect to the updated policy $\pi$.
> > >
> > > We adopt this formulation of value functions of tasks from the prior work on teacher-student curriculum learning [1, 2]. In both [1] and [2], the value functions are defined only on the task or the task parameter, without using any policy parameter or trajectory as inputs. In our work, we follow their practice and extend their definition of value functions for APT-Gen. We do appreciate your suggestions and we also agree that this idea of using trajectory samples to represent tasks can indeed be potentially helpful in some scenarios, but it is not required for the adopted problem formulation of teacher-student curriculum learning. For the aforementioned reasons, the current definitions of $V_1(w; \theta_1)$ and $V_2(w; \theta_2)$ can work sufficiently for APT-Gen and other similar teacher-student curriculum learning frameworks without additional inputs.
> > >
> > > We hope this further clarification has addressed your remaining question. Please don’t hesitate to let us know if you have any additional comments.
> > >
> > > [1] Matiisen et al. “Teacher-Student Curriculum Learning.” In arXiv, 2017.
> > >
> > > [2] Portelas et al. “Teacher Algorithms for Curriculum Learning of Deep RL in Continuously Parameterized Environments.” In CoRL, 2019.

---

> > > > ### Comment · AnonReviewer4 · 2020-11-24
> > > > **Response**
> > > >
> > > > Thank you for your further explanations. I think this point is now reasonable for me. Since $V_1(w; \theta_1)$ and $V_2(w; \theta_2)$ are able to fit the trajectory generated by $\pi$, so $\theta_1, \theta_2$ will incorporate policy information, which further make the loss $L_g$ contains policy information.
> > > >
> > > > As you said in the paper, you want the generated task *indistinguishable from the perspective of the policy*, so that's why I feel it's weird to have your $L_g/V_1/V_2$ without trajectories as input. But now I think this solution also makes sense.
> > > >
> > > > I'll increase my score to 6.

---

> > > > > ### Author Response · Authors · 2020-11-25
> > > > > **Thank you**
> > > > >
> > > > > Dear Reviewer 4,
> > > > >
> > > > > It is great to hear that all of your concerns are now addressed. Thank you again for your constructive suggestions and discussions.

---

### Author Response · Authors · 2020-11-16
**Our General Response**


We thank all reviewers for the constructive suggestions and comments. We appreciate the reviewers' recognition of our technical novelty, clarity, and experimental results. We have submitted an updated version of the paper based on these comments and included additional experiments. Please see the updated file and our responses below for details.

We have updated our paper to include the following changes:

1\) We have included an additional experiment on the influence of the minimum expected return $\delta$ in Sec. D and we have polished our explanations in Sec. B.2 to avoid confusion.

2\) We have updated the notation of the optimization problem in Sec. 3.2. based on Reviewer 4’s suggestions.

3\) We have included discussions on our limitations and future directions in Sec. E.

4\) We have cited and discussed the suggested related work in Sec. 2.

5\) We have revised Sec. 2 and included additional explanations about the variations and the parameterizations of the task spaces in Sec. 1, Sec. 4.2, and Sec. A.

6\) We have added more explanations to justify our selection of baselines.

Please don’t hesitate to let us know for any additional comments on the paper or on the changes.

---

> ### Author Response · Authors · 2020-11-23
> **Our General Response Cont.**
>
> We thank all reviewers for the comments. Based on the last revision, we have further updated the paper to include more changes:
>
> - We have included additional explanations in Sec. 3 to highlight that $V_1$ and $V_2$ are trained to fit the expected values over the distribution of $\tau$, using data collected by the learned policy $\pi$.
> - We have added more detailed comparisons in Sec. 2 about the different goals and problem formulations between POET and APT-Gen.
> - We have added more analysis on the ablation study in Sec. 4.3.
>
> We hope our replies and updates in this and the last revision have addressed all your previous questions. We would be happy to provide further clarifications/revisions, if you have any remaining questions before the discussion period ends.

---

### Decision · Program_Chairs · 2021-01-07
**Final Decision**

**Decision:**

Accept (Poster)

**Comment:**

I thank the authors for their submission and very active participation in the author response period. The paper is well written [R3,R4], tackles a hard problem [R4] in a novel way [R4] with interesting and convincing results [R2]. R3 noted that an empirical comparison to POET would be appropriate. However, in my view the authors addressed these concerns in a satisfactory manner. It seems that R3 has not updated their assessment nor confirmed their current score based on the author response. I am therefore discounting the only review voting for rejection and am siding with R1, R2 and R4. Thus, I recommend acceptance.